# Prebiotic Treatment During Pregnancy Prevents Social Deficits Associated with Autism Spectrum Disorder-like Behavior Induced by Maternal Immune Activation

**DOI:** 10.3390/microorganisms14010060

**Published:** 2025-12-26

**Authors:** Frederico Rogério Ferreira, Guilherme Araujo Rouvier, Lucas Hassib, Raphael de Andrade Morraye, Yago Soares Pereira da Silva, Luís Fernando Saldanha da Gama, Iuri Souza Pereira, Angelica Oliveira Gomes, Maria Julia Granero Rosa, Ana Carolina de Morais-Scussel, Greice Andreotti de Molfetta, Wilson Araujo Silva

**Affiliations:** 1Oswaldo Cruz Foundation, Oswaldo Cruz Institute, Rio de Janeiro 21040-360, RJ, Brazil; guilhermerouvier@gmail.com (G.A.R.); lucas.hassib@usp.br (L.H.); yagosoares39@edu.unirio.br (Y.S.P.d.S.); luisfernandogama04@gmail.com (L.F.S.d.G.); iurdboitec@outlook.com (I.S.P.); 2Research Group in Neurodevelopmental Disorders, Oswaldo Cruz Institute—IOC, Research Center, Innovation and Vigilance in COVID-19 and Sanitary Emergencies, Av. Brasil, Rio de Janeiro 4365, RJ, Brazil; 3Department of Pharmacology, Ribeirão Preto Medical School, University of São Paulo, São Paulo 14049-900, SP, Brazil; 4Laboratory of Molecular Genetics and Omics, Genetics Department, Ribeirão Preto Medical School, University of São Paulo, Ribeirao Preto 14049-900, SP, Brazil; morraye@usp.br (R.d.A.M.); gamolf@fmrp.usp.br (G.A.d.M.); wilsonjr@usp.br (W.A.S.J.); 5Laboratory of Cell Interaction, Federal University of Triângulo Mineiro, Uberaba 38025-180, MG, Brazil; angelica.gomes@uftm.edu.br (A.O.G.); mariajuliagranero@gmail.com (M.J.G.R.); ana.morais@uftm.edu.br (A.C.d.M.-S.)

**Keywords:** autism spectrum disorder, maternal immune activation, *Toxoplasma gondii*, inulin, gut microbiome

## Abstract

Maternal exposure to infectious agents has been associated with an increased risk of mental disorders in offspring, such as autism spectrum disorder. Evidence suggests that maternal immune responses during infection can significantly impact the neurodevelopment of the offspring, potentially affecting central nervous system functions in the future. Inulin is an indigestible soluble fiber that acts as a prebiotic. It promotes the production of short-chain fatty acids, such as butyrate, which can help inhibit the production of pro-inflammatory cytokines. Thus, this study aims to investigate whether inulin treatment during pregnancy can mitigate or reduce the impact of maternal immune activation (MIA) on the neurodevelopment of the offspring. Swiss mice were used in a dose–response study to evaluate the protective effects of inulin against maternal exposure to soluble *Toxoplasma gondii* antigen. Adult offspring of both sexes underwent behavioral assessments, and their gut microbiota was characterized. Both males and females in the soluble *T. gondii* antigen (STAg) group exhibited reduced sociability, as evidenced by the three-chamber social interaction test. Moreover, co-treatment with inulin mitigated this effect. Additionally, anhedonia was observed only in female offspring from the MIA group, but treatment with 1% and 3% inulin also mitigated this effect. The analysis of fecal microbiota showed significant differences between the STAg and inulin treatments at both the family and genus levels. Therefore, inulin appears to have a potential protective effect on the neurodevelopment of the offspring exposed to maternal antigenic challenges during pregnancy mediated by offspring microbiome modulations.

## 1. Introduction

Autism spectrum disorder (ASD) is a complex neurodevelopmental condition characterized by deficits in communication, abnormal social interactions, and restrictive and/or repetitive interests and behaviors [1]. Although sociability deficits are core symptoms of ASD, a variety of other characteristics, including aggression, hyperfocus, epilepsy, and cognitive impairments, may also be present, varying in severity across the spectrum [2,3,4]. The prevalence of ASD has risen globally over the past few decades [5,6,7], and it is considered a serious public health issue. A 2022 meta-analysis revealed that the prevalence of ASD is 0.4% in Asia, 1% in the Americas, 0.5% in Europe, 1% in Africa, and 1.7% in Australia [8].

Although genetic factors clearly contribute to the risk of ASD [9], environmental factors are also involved. Studies have indicated a significant connection between neurodevelopmental changes occurring in the early stages of brain formation, specifically during gestation, and the development of various neuropsychiatric disorders, such as ASD [10]. For example, previous studies have linked prenatal infection with rubella and cytomegalovirus to the development of ASD in offspring [11,12]. However, the mechanisms associating maternal infections with mental disorders are still poorly understood. There is evidence that maternal infections can disrupt the unique immune balance that occurs during pregnancy, leading to a temporary positive regulation of pro-inflammatory cytokines, such as IL-6, IL-8, and TNF, which may influence fetal neurodevelopment [13,14,15,16,17]. Furthermore, a meta-analysis conducted by Jiang and colleagues indicated that maternal infections during pregnancy, whether bacterial, viral, or caused by other pathogens, are associated with a 12% increase in the risk of ASD in offspring [18].

Therefore, the neurodevelopmental mechanisms associated with ASD have been investigated using animal models of gestational immune stress. In this context, preclinical models have demonstrated that maternal immune activation (MIA) disrupts neurodevelopment, leading to depressive- and anxiety-like behaviors, repetitive actions, and reduced sociability in offspring [19]. Indeed, multiple immunological agents, such as poly I:C [20] and lipopolysaccharide (LPS) [21], are used in MIA models to induce neurodevelopmental impairments by mimicking, in some aspects, the imbalance of immune mediators observed in maternal infections. Regarding *Toxoplasma gondii*, its relationship with ASD remains debated [22]. Indeed, clinical studies support the hypothesis that dysfunctions in CD4^+^ T lymphocyte subpopulations, specifically, an increase in Th17 cells and a decrease in Tregs, may contribute to neurodevelopmental impairments in ASD [10,23]. A mouse model of MIA using soluble *T. gondii* antigen (STAg) has shown that it can induce a pro-inflammatory immune response, characterized by the upregulation of cytokines such as interleukin-6 (IL-6) in brain astrocytes and a pro-inflammatory T-cell profile in the periphery [24]. This model demonstrated that the offspring of mice exposed to STAg exhibited autism-relevant behaviors and abnormal brain microstructure. Additionally, the presence of *T. gondii* antigens can lead to the expression of IL-17 by CD4^+^ and CD8^+^ T lymphocytes, which plays a role in the inflammatory response to the parasite [25]. Additionally, the model of MIA can reproduce specific adaptive cellular immune responses resembling the CD4^+^ T cell profile observed in autistic children [26,27,28].

Moreover, a growing body of evidence has shown the influence of the enteric system on the etiology of mental disorders, through a pathway described as the microbiota–gut–brain axis [29,30]. This axis plays a significant role in regulating brain function, particularly in emotional and behavioral processes, as well as in modulating stress responses related to the hypothalamic–pituitary–adrenal axis [29,31]. Research indicates that the composition of the gut microbiota is linked to mental disorders because it influences the bioavailability of neuroactive substances, formation of inflammasomes, production of pro-inflammatory serum modulators, and other alterations in the production of immune system-related metabolites, such as short-chain fatty acids (SCFAs) [30,32,33]. For example, there is evidence of altered inflammatory states, particularly notable in disorders like ASD, which are accompanied by changes in the microbiota, suggesting a significant role of gastrointestinal system alterations in neuropsychiatric disorders [34]. Furthermore, several studies have indicated maternal dysbiosis as a significant environmental factor in the etiology of mental disorders in offspring, as observed in the meta-analysis conducted by Hassib and colleagues, which showed that alterations in maternal gut microbiota during pregnancy lead to changes in sociability and stereotyped behaviors related to psychiatric disorders in offspring [35]. Thus, many pieces of evidence suggest that intestinal microorganisms are important for the etiology of ASD.

Prebiotics are non-digestible food ingredients that selectively stimulate the growth of beneficial bacteria and promote the activity of a limited number of health-promoting bacteria, thus benefiting the host [36,37]. Inulin, specifically, is a prebiotic that is digested by the gut microbiota, producing large quantities of short-chain fatty acids (SCFAs), such as acetate, propionate, and butyrate [38,39]. Evidence indicates that butyrate acts directly as an anti-inflammatory agent by deactivating the NF-κB intracellular transcription factor pathway, consequently attenuating the synthesis of inflammatory cytokines [40]. Together, these findings support the hypothesis that using inulin may help mitigate the negative effects of MIA on offspring by favoring bacteria that digest this fiber and produce higher levels of butyrate. With the increase in colonic butyrate, maternal inflammatory processes may be alleviated, thus reducing their impact on offspring neurogenesis.

## 2. Objective

To explore whether treatment with the prebiotic inulin can alleviate or prevent behavioral changes induced by immunogenic stress from immune activation with *T. gondii* antigens during pregnancy in mice.

## 3. Methods

### 3.1. Animals and Experimental Design

Male and female Swiss mice of 6 weeks of age were provided by the Animal Facility of the Institute of Science and Technology in Biomodels (Rio de Janeiro, Brazil) (ICTB/Fiocruz). The experiments were conducted at the Leonidas/IOC Animal Facility (Bioterium, Rio de Janeiro, Brazil) under a 12:12 h light/dark cycle, at a controlled temperature of 22–24 °C, in micro-isolation chambers. Food, water, and all equipment used for animal handling were sterilized by autoclaving before being introduced into the animal facility. The animals had free access to a standard chow pellet diet composed of 22% protein, 7% fiber, and 4% fat (Nuvilab, Curitiba, Brazil; cat# 1001110036). Experiments were conducted between 8:00 a.m. and 2:00 p.m. All procedures followed local and international standards for handling and care of experimental animals and were previously approved by the local committee of ethics in research with laboratory animals (CEUA L-001/2017-A2).

After a one-week acclimation period at the experimental bioterium, an animal facility assistant, blinded to the experimental procedures, randomly placed the males and females, mated at a ratio of 1:3, until the observation of a vaginal plug, which was considered as gestation day 0.5. Females were isolated, and their weight was monitored daily to track gestational progression. Upon confirmation of pregnancy, the assistant isolated the dams to be randomly assigned to the experimental groups (5–6 animals per group). The power analysis for the behavioral assessment was conducted using the social preference test, based on previous results from our group and on literature regarding social behavior in inbred litters, assuming a between-group difference of Δ = 20 units (1 SD), to achieve 80% power at a two-sided α = 0.05 [19,41]. Because offspring within a litter are correlated, the litter (dam) was treated as the experimental unit. Considering m = 6 offspring measured per litter and an intra-litter intraclass correlation (ICC) of 0.40, yielding a design effect DE = 1 + (m − 1) × ICC = 3. Under these assumptions and SD = 20, the required number of litters per group was eight. These groups consisted of the control (Vehicle + Vehicle), STAG immune-stimulated females treated with a vehicle for inulin (STAg + Vehicle), and STAG immune-stimulated females treated with varying concentrations of inulin: 1% (STAg + Inu 1%), 3% (STAg + Inu 3%), and 10% (STAg + Inu 10%). At birth on embryonic day 21.5 (E21.5), each pup remained with its mother until postnatal day 21 (P21), after which they were group-housed with same-sex littermates (maximum of four per cage). Finally, upon reaching adulthood (P55), the fecal samples were collected, and the animals were evaluated using four behavioral tests in the following order: three-chamber social interaction (SI), elevated plus maze (EPM), sucrose spray (SP), and marble burying (MB). The tests were recorded and later evaluated as blind tests for the experimental groups. Following the behavioral tests, the animals were euthanized (Figure 1). For the 16s sequencing and microbiome analyst, each fecal sample was codified by number and processed by an experimenter blinded to the group. The code was revealed only after sequencing, before uploading to the MicrobiomeAnalyst 2.0 platform.

### 3.2. Treatments

#### Immune Activation with STAg Antigen and Inulin Treatment

The STAg or PBS (vehicle) was used to activate the maternal immune response during the gestational period. On E16.5, pregnant female mice were subcutaneously injected with a single dose of 0.06 mg/Kg of STAg solution at concentration (30 mg/L) or PBS, s.c. injection in a volume of 2 µL/g animal, following a protocol with minor modifications [19]. These modifications reduced miscarriages by STAg administration. The embryonic day for MIA manipulation was chosen due to its primary effects on key developmental processes, including neurogenesis, oligodendrogenesis, axonal growth, and interneuron migration (for more details, see Thion & Garel, 2017 [42]).

According to this study, earlier exposure may be more associated with mood disorders, while later exposure is more likely to involve neurodevelopmental disorders, such as schizophrenia and ASD, as it primarily affects key developmental processes, including neurogenesis, oligodendrogenesis, axonal growth, and interneuron migration (for more details, see (Thion & Garel, 2017 [42]). We have included this information in Section 3.

Inulin (Metachem Ind. e Com. S/A, São Paulo, Brazil) was used as a treatment in pregnant females. Upon reaching embryonic day 3.5 (E3.5), daily treatment with inulin was initiated until the birth of the offspring. Inulin dissolved in drinking water was administered orally via gavage at concentrations of 1%, 3%, and 10%, with administration of regular drinking water for the control group, at a total volume of 100 µL per animal. The treatment scheme was based on previous studies reporting either a reduction of neurotoxicity in offspring from dams exposed to the rotenone model [43] or protective effects on social behaviors in the valproic acid mouse model of ASD, accompanied by increased fecal levels of *L. reuteri* [44].

### 3.3. Behavioral Tests

#### 3.3.1. Three-Chamber Social Interaction

The three-chamber social interaction (SI) test for sociability and preference for social novelty was conducted according to a previously described adapted model [45]. The task is carried out in two stages. In the first stage, the animal’s preference is evaluated by measuring the time spent or exploratory activity (interactions) with another animal compared to an object. In the second stage, called the novelty sociability test, the evaluation focuses on whether the animal prefers to explore an unfamiliar animal over a familiar one. The mice were first habituated to the arena for 5 min, which was a rectangular acrylic arena (45 × 15 × 15 cm), divided into three sections with passages allowing communication between them. In the second 10 min section, sociability was measured. During this period, mice could interact through an acrylic wall with an object contained in one chamber or with a strange animal (Stranger 1) of the same sex and age in the other chamber. The time spent in the interaction zone, defined as the cumulative time spent in the chamber containing either the object or the stranger animal (Stranger 1), and the number of interactions, counted based on sniffing, touching, or facing the stimuli, were recorded and evaluated. Finally, preference for social novelty was assessed during a third 10-min session by replacing the object with a second unfamiliar mouse (Stranger 2) in the chamber. The time spent in the interaction zone and the number of interactions with Stranger 1 or Stranger 2 were recorded and evaluated as described above.

#### 3.3.2. Sucrose Spray

The sucrose spray (SS) test was conducted following the model of Frisbee [46]. The mice were placed in a clean acrylic box (20 × 30 × 13 cm) after the application of a 10% sucrose solution to the nape. The animals were recorded for 5 min, and facial grooming behavior was assessed. Animals that do not respond to the sweet stimulus of a sucrose spray applied dorsally may be interpreted as displaying anhedonic-like behavior, like the loss of interest in previously pleasurable activities observed in patients with major depressive disorder.

#### 3.3.3. Elevated Plus Maze

The elevated plus maze (EPM) was conducted according to a model previously described [47]. The test consisted of an acrylic apparatus, elevated 45 cm above the floor, with two open arms of 15 cm and two enclosed arms of 15 cm, arranged in a cross shape. The mice were placed in the central square of the maze facing a closed arm and allowed to explore for 5 min. The mice’s exploration behavior and risk assessment were recorded and evaluated. The percentage of time spent in the open arms [(open/total) × 100] and the total number of arm crossings were recorded. Risk assessment was evaluated by counting the frequency with which the animal exhibited a downward head movement toward the floor from the open arms while keeping at least two-thirds of its body in the closed (protected) arm. The reduction of exploratory activity in the open arms (unprotected areas) or an increased number of risk-assessment behaviors may be interpreted as generalized anxiogenic-like behavior.

#### 3.3.4. Marble Burying

The marble burying (MB) test was conducted following a model previously described [48]. Each individual mouse was placed in a clean transparent cage (20 × 30 × 13 cm) filled with 5 cm of bedding containing 20 transparent glass marbles, which were evenly spaced on the surface of the bedding in a 5 × 4 arrangement. After 30 min, the mice were carefully removed from the test cage, and the number of marbles buried (up to two-thirds of their depth covered with bedding) was counted. An increased number of marbles buried is interpreted as repetitive/compulsive-like and stereotyped-like behavior.

### 3.4. 16s Sequencing of the Animal Microbiome

The fecal samples were collected from the animals’ housing cages at P55 before behavior tests. Each 16S sequencing was performed on a pool of 3–4 adult male animals from the same cage and experimental group. These samples were stored in 95% alcohol and kept at −80 °C. DNA extraction was performed using the QIAmp PowerFecal Pro DNA kit (QIAGEN, Germantown, MD, USA) following the manufacturer’s instructions. The V4 region of the 16S ribosomal RNA gene was amplified by PCR using the primer sequences 515F (5′-TCGTCGGCAGCGTCAGATGTGTATAAGAGACAGGTGCCAGCMGCCGCGGTAA-3′) and 806R (5′-GTCTCGTGGGCTCGGAGATGTGTATAAGAGACAGGGACTACHVGGGTWTCTAAT-3′).

For library construction, the Nextera XT DNA Library Preparation Kit (Illumina, San Diego, CA, USA) was used, incorporating Illumina adapters and molecular barcodes. The amplicons were labeled for subsequent sequencing on the MiSeq platform (Illumina) using the MiSeq v2 Reagent Kit (500 cycles) for 2 × 250 paired-end reads [49]. The sequences obtained in FastQ format were processed using the QIIME2 pipeline (version 2020.2) for alignment and demultiplexing [50]. Quality control was performed on the demultiplexed sequences using the RCPP package of the DADA2 software, version 1.28.0, to filter chimeric sequences [51]. Amplicon sequence variants (ASVs) from the V4 region were assigned taxonomy using the assignTaxonomy function from the DADA2 package, utilizing the SILVA reference database. We adjusted the minBoot parameter to 85, meaning that only taxonomic assignments with a bootstrap confidence of at least 85% were accepted. By default, minBoot is set to 50, but we increased this threshold to enhance the reliability of taxonomic classifications, reducing the likelihood of incorrect assignments at lower confidence levels. The classification process in DADA2 involves a k-mer-based naive Bayesian classifier, which matches ASVs to the closest reference sequences in the SILVA database [52]. Visualization and statistical analyses of the microbiome data were performed using the MicrobiomeAnalyst platform [53].

### 3.5. Statistical Analysis

All statistical analyses were performed using GraphPad Prism 8.0.2 software. The data were initially assessed for normality using the Shapiro–Wilk test (*α* < 0.05). When normality was confirmed, data obtained from the SI test were analyzed using two-way ANOVA. Subsequently, the Sidak post hoc test was employed for multiple comparisons. For the EMP, SS, and MB tests, only treatment was used as a factor, analyzed by one-way ANOVA followed by Sidak’s post hoc testing. Analyses in which the samples did not meet the requirements of normal distribution and/or homogeneity of variance were conducted using the Kruskal–Wallis test, followed by the Dunn multiple comparisons test. The significance level for all analyses was set at *p* < 0.05. To avoid pseudo-replication, linear mixed-effects models with a random intercept for litter (dam) were applied to account for intra-litter correlation; fixed effects included STAg and inulin (treatments), and degrees of freedom were estimated using the Satterthwaite method.

The microbiota analyses were conducted using the MicrobiomeAnalyst platform, with alpha diversity evaluated through Student’s *t*-test and ANOVA. Subsequent pairwise comparisons between groups were conducted using a paired *t*-test for matched samples and the Mann–Whitney test for independent samples. For beta diversity, multiple comparisons were performed using PERMANOVA analysis. Finally, relative abundance was assessed using the DESeq2 statistical method [54]. The significance level for all analyses was set at *p* < 0.05. Additionally, the multi-testing adjustment was performed using the Benjamini–Hochberg procedure (FDR).

## 4. Results

### 4.1. Effects of Maternal Inulin Treatment on Social-like Behavior Deficits Induced in the MIA Mouse Model with STAg, Assessed Using the Three-Chamber Social Interaction

Linear mixed-effects models with litter (dam) or sex as random intercepts and treatments (STAg and inulin) as fixed effects indicated that sex (F_4,91_ = 0.499; *p* = 0.776) and litter (F_10,91_ = 1.851; *p* = 0.056) did not interact with treatments when evaluating the sociability parameter (% time spent in chamber 2). For the number of interactions parameter, the linear mixed-effects models also reveal that litter (F_10,91_ = 1.222; *p* = 0.283) or sex (F_4,91_ = 0.544; *p* = 0.742) did not interact with the treatments.

The two-way analysis of variance (ANOVA) revealed that male offspring exhibited a preference to spend time on the site of Stranger 1 mice compared to the object site (F_1,54_ = 31.70; *p* < 0.0001), independently of MIA or inulin treatments. Moreover, the Sidak post hoc analysis showed that the preference to interact with a novel mouse rather than an object was not statistically significant for the male offspring born to mothers exposed to STAg (*p* = 0.2275), suggesting that prenatal exposure to STAg affects the sociability of these animals. Interestingly, maternal treatment with 10% inulin successfully restored the preference of offspring for interactions with the Stranger 1 (61.37 ± 2.14) compared to the object (38.62 ± 2.14; *p* = 0.0385), similar to that observed for the control group, where the offspring spent more time exploring the chamber with the Stranger 1 (62.27 ± 2.34) compared to the time spent in the chamber with the object (37.73 ± 2.34; *p* = 0.0030). The treatment of mothers with 1% and 3% inulin was not effective in mitigating the effects of STAg (Figure 2a).

For the number of interactions parameter, male offspring also exhibited a preference to interact with Stranger 1 mice compared to the object site (F_1,54_ = 75.80; *p* < 0.0001), with no interaction with STAg exposure or inulin treatment (*p* = 0.3848; *p* = 0.0953, respectively). However, maternal exposure to STAg also significantly reduced the number of interactions with the Stranger 1 (39.79 ± 4.67) compared to the object (28.54 ± 2.53; *p* = 0.0928), in contrast with the control group that engaged in more interactions with the Stranger 1 (42.92 ± 4.13) compared to the object (20.42 ± 2.14; *p* = 0.0001). Moreover, offspring from mothers treated with 1%, 3%, and 10% inulin demonstrated a preference for interaction with the Stranger 1 (38.57 ± 5.12; 50.08 ± 5.19; 42.00 ± 4.24, respectively) compared to the object (16.86 ± 1.30; *p* < 0.0001; 27.33 ± 2.46; *p* < 0.0001; 23.25 ± 3.08; *p* = 0.0129, respectively), despite being exposed to MIA with STAg during gestation (Figure 2b).

The two-way ANOVA revealed that female offspring exhibited a significant preference for time spent in the Stranger 1 compartment compared to the object (F_1,49_ = 35.36; *p* < 0.0001), with a marginal effect of interaction with STAg exposure (*p* = 0.0548) and no effect of interaction with the inulin treatment (*p* = 0.2949). Sidak’s post hoc analyses revealed that animals from the control group spent more time exploring the chamber containing Stranger 1 (58.75 ± 2.25) compared to the object chamber (41.25 ± 2.25; *p* = 0.0027). In contrast, animals exposed to STAg during pregnancy exhibited a reduced preference for spending time in the chamber with Stranger 1 (54.41 ± 1.91) compared to the object chamber (45.59 ± 1.91; *p* = 0.3365). Maternal treatments with inulin at doses of 1% or 10% prevented the effects of MIA, increasing the preference to spend time in the chamber with stranger 1 (59.02 ± 3.00; 59.58 ± 3.14, respectively) rather than the object chamber (40.98 ± 3.00; *p* = 0.0047; 40.41 ± 3.14; *p* = 0.0125, respectively; Figure 2c).

Furthermore, when assessing the number of interactions, ANOVA identified significant effects between Stranger 1 and the object (F_1,49_ = 52.75; *p* < 0.0001), and interaction with STAg exposure (F_4,49_ = 2.78; *p* = 0.0369), but not interaction for inulin treatments (*p* = 0.3017). The post hoc test revealed that female offspring from the control group engaged in more interactions with Stranger 1 (41.54 ± 5.20) compared to the object (21.31 ± 2.84; *p* = 0.0033). The female offspring subjected by gestational exposure to immunogenic challenge with STAg does not exhibit preference to interact with Stranger 1 (38.08 ± 3.99) than the object (25.92 ± 2.71; *p* = 0.1882), indicating that MIA also impairs sociability behavior. Additionally, maternal treatments with 1% and 3% inulin reversed the effects of STAg in these offspring, increasing the preference to interact with Stranger 1 (51.54 ± 6.78; 51.90 ± 6.35, respectively) compared to the object (21.09 ± 2.07; *p* < 0.0001; 31.20 ± 2.97; *p* = 0.0100, respectively). Finally, prenatal treatment with 10% inulin does not significantly modulate the preference to interact with the Stranger 1 animal (*p* = 0.0980; Figure 2d). Collectively, these results indicate that MIA with STAg may induce a reduction in sociability in female offspring, as well as in male adults, while the use of inulin during gestation prevents these behavioral impairments.

In assessing social novelty preference in males, the linear mixed-effects models also indicated that litter (F_10,91_ = 1.108; *p* = 0.365) and sex (F_4,91_ = 1.984; *p* = 0.088) did not interact with treatments when evaluating the sociability parameter. For the number of interactions parameter, the linear mixed-effects models also reveal that litter (F_10,91_ = 0.471; *p* = 0.471) or sex (F_4,91_ = 2.015; *p* = 0.084) did not interact with the treatments. The two-way ANOVA revealed that animals exhibited a preference for spending time exploring the compartment of Stranger 2 rather than Stranger 1 (F_1,54_ = 10.70; *p* = 0.0018). For this parameter, ANOVA indicates a significant main effect of STAg immunostimulation (F_4,54_ = 427.2; *p* < 0.0001), with a trend toward significance for the interaction between STAg exposure and inulin treatments (F_4,54_ = 2.222; *p* = 0.0787). Although the animals in the control group spent more time in the chamber with Stranger 2 (55.75 ± 4.12) compared to Stranger 1’s chamber (44.25 ± 4.12), this difference was not statistically significant (*p* = 0.3126). The Sidak post hoc analyses revealed that maternal treatments with 1% and 3% inulin increased the preference for social novelty, as indicated by interaction times with Stranger 2 and Stranger 1 (59.12 ± 2.28 versus 40.88 ± 2.28; *p* = 0.0136; 59.33 ± 3.62 versus 40.67 ± 3.62; *p* = 0.0218, respectively). Additionally, the animals from mothers exposed to STAg and treated with 10% inulin do not exhibit a preference for spending time in the chamber with the unfamiliar stranger (*p* = 0.956; Figure 3a).

For the total number of interactions, ANOVA revealed a significant preference for the tested mice to interact with Stranger 2 rather than Stranger 1 (F_1,54_ = 22.16; *p* < 0.0001). There was also a significant effect for the treatments (F_4,54_ = 3.993; *p* = 0.0066), as well as for the interactions between STAg exposure and inulin treatments (F_4,54_ = 3.813; *p* = 0.0084). Post hoc tests suggested that male offspring from the control group engaged in more interactions with Stranger 2 (31.83 ± 4.26) compared to Stranger 1 (21.9 2 ± 2.38; *p* = 0.0365), indicating a preference for new social interactions. However, prenatal exposure to STAg abolished the preference for social novelty, as indicated by the absence of a significant difference in interaction time with Stranger 2 versus Stranger 1 (34.61 ± 2.76 vs. 34.61 ± 2.41; *p* > 0.9999). Prenatal treatment with 1% or 3% inulin reversed the effects of STAg, increasing interactions with Stranger 2 (29.64 ± 3.04 and 42.00 ± 3.24, respectively) compared to Stranger 1 (18.93 ± 2.79; *p* = 0.0100; 25.08 ± 2.57; *p* < 0.0001, respectively; Figure 3b).

In the social novelty test, a two-way ANOVA indicates that female offspring may not exhibit a preference for spending time in the interaction chamber with the unfamiliar mice (F_4,49_ = 2.422; *p* = 0.1261). However, maternal exposure to STAg antigen may have a modulatory effect on this behavior (F_4,49_ = 2.91; *p* = 0.0309), with a marginal interaction with the inulin treatment (F_4,49_ = 2.495; *p* = 0.0548). However, the post-test did not detect any differences in the time spent in the interaction chamber with the familiar and unfamiliar animals in this group (Figure 3c). For number of interactions parameter, ANOVA did not identify significant effects between Stranger 2 and Stranger 1 (F_1,49_ = 1.998; *p* = 0.1638), nor significant differences between groups (F_4,49_ = 1.94; *p* = 0.1192) or interactions between maternal treatments with STAg and inulin. Only the offspring from the control group exhibited a preference to interact with Stranger 2 (34.00 ± 2.87) compared to Stranger 1 (24.38 ± 3.01; *p* = 0.0462), as indicated by the post hoc analysis. However, maternal treatment with inulin at any dose did not modulate the social novelty preference behavior when evaluated by this parameter (Figure 3d).

Therefore, these data suggest that maternal STAg exposure reproduces social impairments in the mouse model of MIA in both experimental groups—male and female offspring. Additionally, prenatal treatment with the prebiotic inulin may protect against social deficits induced by immunological stress from STAg.

### 4.2. Effects of Maternal Inulin Treatment on Depressive-like Behavior Induced in the MIA Mouse Model with STAg, Assessed Using Sucrose Spray Test

The effect of prenatal inulin treatment on depressive-like behavior associated with MIA was also investigated. Maternal exposure to the STAg antigen had no effect on grooming behavior (*p* = 0.689), nor was there an effect of inulin treatment (*p* = 0.844) or sex (*p* = 0.129) when evaluated by latency parameter (Figure 4a,c). Analysis of the grooming frequency parameter using the generalized linear model revealed that grooming frequency was significantly modulated by STAg exposure (*p* < 0.001) and inulin treatment (*p* = 0.047), with no effect of sex (*p* = 0.201). Moreover, the linear mixed-effects models suggest that litter did not interact with treatments when evaluated by the latency for first grooming (F_5,112_ = 0.864; *p* = 0.577) or number of grooming (F_5,112_ = 0.499; *p* = 0.776). Similarly, the latency (F_5,112_ = 2.075; *p* = 0.073) and number grooming (F_5,112_ = 1.218; *p* = 0.305) did not interact with the animals’ sex. The Kruskal–Wallis post-test suggests that maternal exposure to STAg induced a significant reduction in the number of grooming episodes compared to the control group (6.00 ± 1.07 versus 21.15 ± 3.34; *p* = 0.0046) in female offspring. Additionally, prenatal treatment with 1% inulin (12.18 ± 2.81; *p* = 0.7649) and 3% inulin (16.93 ± 4.76; *p* = 0.6455) prevented the effects of MIA on grooming behavior in this group, as shown in Figure 4d. The prenatal supplementation with a 10% inulin solution failed to restore the grooming behavior when compared to the control group (2.85 ± 0.74; *p* = 0.0003). Collectively, this data indicates that maternal immunogenic stress induced by exposure to STAg can lead to depressive-like symptoms in adult offspring, while treatment with the prebiotic inulin may provide benefits in mitigating the effects of MIA. However, under these experimental conditions, this effect was more pronounced in female offspring.

### 4.3. Effects of Maternal Inulin Treatment on Anxiety-like Behavior and Repetitive/Compulsive-like Behavior or Stereotyped-like Behavior Induced in the MIA Mouse Model with STAg, Assessed Using Elevated Plus Maze and Marble Burying Test

The EPM test was employed to investigate anxiety-like behavior in offspring from mothers subjected to MIA, as well as the effects of inulin treatment. Under the experimental conditions of this study, no effects of inulin treatment or STAg exposure were detected in males for the open arms (*p* = 0.0790) or risk assessment (*p* = 0.5815; Appendix A).

In females, no effects of the treatments were observed on the open arms (*p* = 0.1828; Appendix A) or risk assessment (*p* = 0.1978; Appendix A). Additionally, no treatment effect was found on animal locomotion, as evidenced by the number of crossings in males and females (*p* = 0.5690; *p* = 0.5773, respectively; Appendix A), indicating no motor impairment due to the treatments.

Finally, to assess obsessive–compulsive-like behavior, the MB test was applied. The Kruskal–Wallis test found no significant effect on the number of marbles buried by male offspring (*p* = 0.8753) or females (*p* = 0.4922; Appendix A). Thus, neither prenatal treatment with STAg nor co-treatment with inulin could induce an increase in the impulsive and repetitive behavior of the offspring.

### 4.4. Effects of Maternal Inulin Treatment on the Fecal Microbiome of Adult Offspring

The 16S ribosomal RNA extracted from the feces of adult offspring was used to characterize their gut microbiota. All the samples achieved sequencing saturation, as shown by the rarefaction curve (Appendix A), indicating full coverage across all groups for our standardized conditions. Despite the limited number of sequenced samples per group, each sample comprised a set of five animals from the respective group, providing good representativeness for each treatment or intervention.

Both maternal immunogenic challenge with STAg and prenatal supplementation with inulin significantly modulated the alpha diversity at the family level (F_4,9_ = 9.4091; *p* = 0.0151), and also at the genus level (F_4,9_ = 7.875; *p* = 0.0219), as indicated by the Chao1 index. Moreover, maternal supplementation with inulin demonstrates a dose-dependent effect in an inverted U-shaped manner, as assessed by the Chao1 index at both the family and genus levels (Figure 5a,b). The immunogenic stress by STAg during pregnancy increased the alpha diversity when evaluating at genus and family levels in the adult offspring, although it does not reach the significance level (*p* = 0.1577; *p* = 0.0917, respectively). The analysis of alpha diversity using the Shannon index showed a marginal effect of the treatments at the genus level (F_4,9_ = 4.036; *p* = 0.07) but no effect at the family level (F_4,9_ = 3.47; *p* = 0.102; Figure 5c,d). Similarly, alpha diversity analysis using the Simpson index did not detect an effect of the treatments at either the genus (F_4,9_ = 1.42; *p* = 0.34) or family level (F_4,9_ = 1.31; *p* = 0.37; Figure 5e,f).

The analysis of alpha diversity by using the Chao1 index revealed that maternal treatment with inulin 10% significantly reduced the effect of MIA for the genus levels (*p* = 0.013), with a marginal effect at family level (*p* = 0.069), although after applying the Benjamini–Hochberg procedure correction, the statistical significance was no longer observed for both parameters (FDR = 0.1759; FDR = 0.1361, respectively; Figure 5a,b). These results together indicate that the behavioral changes observed in adult offspring related to ASD, resulting from maternal immunogenic stress during pregnancy, may stem from gut microbiota modulation, which is passed from mother to offspring.

When assessing beta diversity using the Bray–Curtis index, both MIA and inulin treatment did not induce significant differences at the family level (*p* = 0.8160; F_4,9_ = 0.6311; Figure 5b). However, at the genus level, both treatments significantly modulated the gut microbiota of the offspring (*p* = 0.0160; F_4,9_ = 3.2414; Figure 5d). However, the post hoc test did not identify these differences in a pairwise comparison. Thus, these findings suggested that maternal prenatal supplementation with inulin, across all doses, may be associated with neuroprotective and behavioral benefits in offspring. However, expanding the sequencing data is essential to gain a more detailed understanding of the microbial communities linked to the beneficial effects of inulin in mitigating MIA. The relative abundance at the family level is shown in Figure 6. However, due to the large number of “not assigned” sequences, we were unable to create a stacked bar chart at the genus level.

Finally, the 16S ribosomal RNA analyses highlighted the relative abundance of the gut microbiota at both family and genus levels in the adult offspring. The maternal immunogenic challenge with STAg antigens significantly increased the relative abundance of the families *Bacteroidaceae*, *Lachnospiraceae*, *Erysipelotrichaceae*, and *Acholeplasmataceae* in adult offspring compared to the control group (*p* < 0.0001; *p* < 0.0001; *p* < 0.0001; *p* = 0.028, respectively; Figure 7a,c,g,h). Additionally, it led to a notable reduction in the family *Oscillospiraceae* (*p* = 0.008; Figure 7f). Also, MIA increased the abundance of the genera *Bacteroides*, *Lachnospiraceae*, and *Turicibacter*, while decreasing the presence of the genus *Alistipes* compared to the control group (*p* < 0.0001; *p* < 0.0001; *p* < 0.0001; *p* < 0.0001, respectively; Figure 7k,l,n,q). Therefore, these findings suggested that behavioral impairments induced by immunogenic stress during pregnancy may be associated with the modulation of specific microorganisms in adult offspring enteric microbiome.

Maternal inulin supplementation also prevented the effects of MIA in the offspring at microbiome level. The 1% inulin supplementation significantly reduced the relative abundance of the *Bacteroidaceae* and *Acholeplasmataceae* families compared to the STAg group (*p* < 0.0001; *p* < 0.0001, respectively; Figure 7a,h), thereby restoring the microbiota profile to levels like those of the control group. Moreover, the 10% inulin supplementation significantly increased the relative abundance of the *Oscillospiraceae* family compared to the STAg group (*p* = 0.019; Figure 7f), bringing its levels back to those observed in the control group. At the genus level, 1% inulin effectively reduced the abundance of *Bacteroides* to levels like those observed in the control group, in contrast to the STAg group (*p* < 0.0001; Figure 7k). Furthermore, the use of 3% inulin decreased the abundance of *Lachnospiraceae* and increased the abundance of *Alistipes* compared to the STAg group (*p* < 0.0001 for both; Figure 7l,n), restoring levels to those found in the control group. Thus, these results suggested that prenatal supplementation with inulin, across all doses, has a protective effect in preventing or reversing the effects of MIA on the gut microbiota of adult offspring. Additionally, the modulation of the offspring’s gut microbiota could be linked to social behavior restoration.

## 5. Discussion

The etiology of ASD involves interactions among distinct neurodevelopmental features, including alterations in brain maturation patterns, connectivity, and neuroanatomy [55]. These processes are regulated throughout brain development—and even into adulthood life—by complex interactions among microbial metabolites, immune modulators, and neural pathways, collectively known as the gut–brain axis (GBA). Therefore, the present study investigates whether improving the maternal enteric microbiome through supplementation with the prebiotic inulin can protect offspring from immunological stress during pregnancy.

The present findings indicate that male offspring born to dams immunostimulated with STAg during pregnancy exhibit social impairments similar to those observed in patients with ASD. Moreover, females derived from the MIA model with STAg also show social impairments. Additionally, exposure to STAg antigen during pregnancy affects sociability, as the animals lose their preference for interacting with an unfamiliar animal over a familiar one. However, maternal treatment with inulin mitigates social interaction deficits in both sexes. Regarding social novelty, the effect of inulin was observed only in male offspring. Female offspring generated by dams subjected to MIA with STAg exhibit alterations in feeding behavior consistent with anhedonia. However, this effect requires further analysis. Finally, the effects of STAg exposure may be associated with social deficits. Its effects on anxiety-like, depressive-like, or obsessive–compulsive-like behaviors would warrant further analysis.

The mechanisms underlying the consequences of immunological stress in ASD are still under debate. However, our findings add evidence that this may be associated with the modulatory effect of the enteric microbiome on the immune system during brain development. Hence, it is plausible that a distinct inflammatory profile is established during gestation following an immunogenic challenge with STAg, which can influence neurodevelopment and the inflammatory status of the offspring, thereby inducing the observed social impairments. Clinical studies have already indicated that microbial dysbiosis in the gastrointestinal (GI) tract is prevalent among individuals with ASD, potentially contributing to behavioral manifestations and correlating with symptom severity [56]. Moreover, longitudinal studies suggest that decreases in beneficial bacteria such as *Bifidobacterium* and *Lactobacillus* correlate with severity of this disorder. Specific alterations in the gut microbiota, such as increased Firmicutes and decreased Bacteroidetes, have been observed in children with ASD, which correlates with behavioral and GI abnormalities [57]. The metabolites like short-chain fatty acids (SCFAs) by these genera may directly regulate neural process of oligodendrocyte function and subsequent myelination and, indirectly, modulate the neuroinflammatory pathways [58]. Palanivelu et al. (2024) also provide insights into the GBA dynamics using an autistic-like rat model, demonstrating significant brain microstructural changes and reduced microbiota diversity, which are associated with metabolic dysregulation and pro-inflammatory responses [59]. Indeed, several events during gestation may disturb the eubiotic balance of the enteric microbiome. The unhealthy maternal microbiota can be transmitted vertically to the offspring, impacting the critical period of brain development [35,60].

Another interesting study showed that maternal “immunological experiences” or repertoire during pregnancy (embryonic stage) shapes the offspring’s immunological maturation and capabilities in adulthood [61]. This seminal work led to the hypothesis that maternal immunological repertoire, acquired through interactions with the enteric microbiome, may influence the offspring’s neurodevelopment, with implications for susceptibility or resilience to psychiatric disorders, including ASD. Additionally, clinical and preclinical studies have provided evidence of the influence of the maternal microbiome on various mental conditions in offspring [35,62]. The exposure to *T. gondii* antigens, including live tachyzoites, has been shown to modulate cytokine production. Indeed, peripheral blood mononuclear cells (PBMCs) from parturient women exposed to live tachyzoites produced various cytokines, including IL-6, IL-10, IL-12, and TNF-α, indicating a complex immune response that can vary depending on the serological status of the individuals [63]. Additionally, the presence of *T. gondii* antigens can lead to the expression of IL-17 by CD4^+^ and CD8^+^ T lymphocytes, which plays a role in the inflammatory response to the parasite [25]. From the perspective of animal models, previous studies have reported that MIA can lead to decreased sociability in adult offspring [24,64]. Models utilizing LPS and Poly I:C to produce the MIA have demonstrated behavioral changes in the offspring that resemble those seen in ASD patients, including social impairments [65,66]. Our MIA model used STAg as a trigger to replicate ASD-related behavioral phenotypes in the offspring, given that the role of *T. gondii* in the etiology of this condition is well-documented in the literature [22,67]. Increased levels of IgG against *T. gondii* are associated with a higher incidence of ASD in humans [67]. Finally, studies conducted by Zhipeng Xu [24] found that administering STAg during pregnancy induces a pro-inflammatory profile in the offspring that persists into adulthood. This profile is characterized by a decrease in regulatory T cells, an increase in peripheral TH1 and TH17 cells, and elevated IL-6 levels in brain regions such as the prefrontal cortex and hippocampus. The imbalance of these CD4 T cells is consistent with the inflammatory profile observed in children with ASD [26,27,28]. Therefore, this last finding corroborates earlier evidence showing that children with ASD exhibit altered immune profiles and functions, characterized by deficits in regulatory T cells and an increase in pro-inflammatory profile [26]. Together, this evidence underscores the importance of early interventions targeting gut microbiota to potentially mitigate neurodevelopmental changes in ASD.

Regarding other psychotic disorders, studies indicate that 14% to 50% of patients diagnosed with ASD have depression, and 40% to 80% have anxiety disorders [68]. However, other factors are associated with the development of these mental disorders in individuals with ASD. Prenatal exposure to infectious agents, dysbiosis, and stress are highly correlated with behavioral alterations such as depression and anxiety [35,69,70,71,72,73]. In the current MIA model using STAg, no changes in the behavior related to depression were observed in male offspring. However, maternal immunogenic stimulation was only able to induce an increase in grooming behavior in females. Anhedonia is a characteristic behavior of major depressive disorder and can be observed in patients with ASD [74,75]. Epidemiological studies demonstrate that depression has a higher prevalence in females, which may vary with age [76,77] or affect women twice as often as men over a lifetime [78,79]. Therefore, our findings support these studies, suggesting that MIA could be a significant sex-dependent factor in the etiology of this mental disorder. However, further analyses are needed to confirm this idea.

When analyzing anxious-like behaviors, the MIA model using STAg was not able to induce anxiety-related behaviors in the offspring evaluated in the LCE test, unlike studies that used Poly IC, LPS, and other forms of STAg administration [80,81,82]. A wide range of protocols use STAg to trigger MIA, with differences in the route of administration, concentration, and the embryonic day chosen to perform MIA. These factors may account for the varying behavioral outcomes observed across studies. Therefore, this study chose to use the subcutaneous (SC) route instead of the intraperitoneal (IP) route used in previous studies [19,24,83]. The s.c. administration has advantages compared to the i.p. route, such as reducing suffering and stress due to the shallow penetration of the needle into the animal’s body. In contrast, i.p. administration involves needle penetration through the skin, muscle, and peritoneum, which poses a potential risk of internal organ damage [84,85]. Therefore, the chosen route of administration reduces the impacts of prenatal stress that could contribute to increased anxious-like behavior in the offspring [86,87,88].

Repetitive/stereotyped behavior is one of the characteristic symptoms of individuals with ASD, alongside communication deficits and social impairments [89,90]. Studies using MIA models with LPS, poly I:C, and STAg antigens have revealed that offspring exhibit increased repetitive/stereotyped behavior, as assessed in the marble burying test [24,89,91,92]. However, it was not possible to observe an increase in repetitive/stereotyped behavior in our MIA model with STAg nor the elevation of impulsive behavior, characteristic of obsessive–compulsive disorder, as assessed in the marble burying test [93,94].

In this study, inulin was used as a prebiotic capable of modulating the gut microbiota of pregnant females, increasing the production of SCFAs, such as butyrate [38,39,40,95]. Previous studies have reported that prebiotics can improve cognitive performance, anxiety, and depression in rodents [29,96]. Additionally, maternal supplementation with these dietary fibers alleviates memory and sociability impairments in the offspring of mothers from a prenatal obesity model [97]. Furthermore, randomized clinical trials have shown that dietary supplementation with prebiotics enhances social and communication skills in children diagnosed with ASD [98,99,100]. Our findings align with these studies, showing that maternal treatment with inulin during pregnancy reverses the sociability impairments in offspring caused by MIA and restores social novelty preference behavior in male offspring.

Currently, numerous preclinical studies are highlighting the benefits of using prebiotics in managing mental disorders related to depression and anxiety [101]. However, clinical trials assessing the efficacy of prebiotics in treating these disorders remain limited [102]. In our MIA model, prenatal maternal treatment with 1% and 3% inulin successfully reduced anhedonic-like behavior in female offspring, suggesting a beneficial effect of inulin against MIA with STAg. However, the use of 10% inulin did not increase the total number of facial groomings performed. The intake of inulin modulates maternal microbiota, influencing the offspring’s microbiome and maintaining high concentrations of SCFA-producing bacteria such as acetate, propionate, and butyrate [38,39,103]. Studies indicate that acetate has a regulatory function on appetite, potentially leading to reduced food intake [104]. Thus, it is possible that the use of 10% inulin modulated the microbiota of the offspring to produce more acetate, inhibiting their appetite and consequently reducing the total number of facial groomings performed.

In the present study, the prenatal immunogenic stress protocol effectively altered the gut microbiota in adult offspring, as previously documented in the literature [82,105]. The administration of STAg antigens resulted in an increase in the *Bacteroidaceae* family and the *Bacteroides* genus in the offspring. Evidence indicates that autistic children have a higher proportion of the Bacteroidetes phylum compared to the Firmicutes phylum, now referred to as *Bacillota* [34,106,107]. Additionally, MIA with STAg antigens decreased the relative abundance of the *Oscillospiraceae* family, part of the *Bacillota* phylum, creating a microbiota profile similar to that observed in children with ASD. However, maternal co-treatment with inulin successfully restored the balance between Bacteroidetes and *Bacillota*, aligning with previous studies that showed a negative correlation between high dietary fiber intake and the enrichment of Bacteroidetes [108,109]. Furthermore, prenatal supplementation with inulin decreased the relative abundance of *Acholeplasmataceae*, a family within the *Tenericutes* phylum, which remained elevated in the STAg group. A study conducted by Avolio and colleagues found that mice receiving feces from autistic children donors showed an increased population of *Tenericutes*. Moreover, these mice exhibited behavioral phenotypes akin to ASD, including reduced sociability, as assessed by the three-chamber social interaction test [110]. However, drawing biological inferences using the microbiome panel from different taxonomic levels such as phylum, family, and genus is a challenging task, due to hight variability and lower taxonomic resolution. These discrepancies can lead to difficulties in comparing findings across studies that utilize different taxonomic resolutions. Moreover, strain-level diversity within a species can impact microbiome functionality, complicating comparisons between studies that may not resolve taxa to the strain level.

These findings indicate that MIA with STAg antigens can vertically affect the offspring’s microbiota, potentially influencing behaviors associated with ASD. Additionally, maternal supplementation with inulin may partially counteract the negative changes in the offspring’s gut microbiota, also contributing to a recovery of behaviors associated with ASD.

## 6. Conclusions

In conclusion, the behavioral results obtained support our hypothesis that inulin has the potential to mitigate the impacts of MIA on offspring neurodevelopment by promoting butyrate-producing bacteria, which in turn reduce inflammation induced by STAg. Additionally, the characterization of the gut microbiota demonstrated the beneficial effects of prenatal maternal supplementation with inulin in mitigating the dysbiosis caused by MIA.

## 7. Limitations

This study identified several limitations concerning the quality of microbiome sample sequencing. These challenges demanded conducting the microbiome analysis using only forward reads, thereby constraining the extent of taxonomic characterization. Consequently, information on more specific taxonomic levels was limited, as reflected by the high number of classifications not assigned at the genus level. To overcome this limitation and obtain a more detailed microbiome analysis, resequencing samples is necessary. This approach will provide a deeper sequencing level, facilitating the identification of taxonomic groups at finer resolutions and offering a more comprehensive understanding of the microbial composition and diversity within the samples.

## Figures and Tables

**Figure 1 microorganisms-14-00060-f001:**
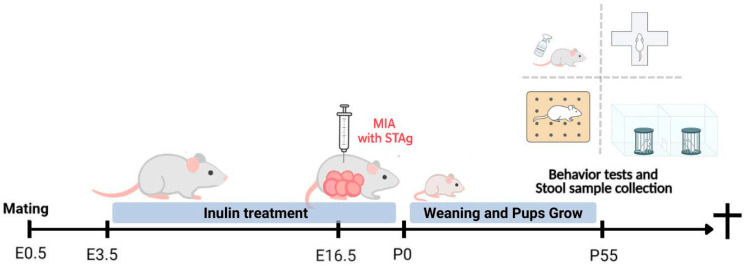
Experimental design. Protocol for MIA, inulin treatment, and behavioral analysis.

**Figure 2 microorganisms-14-00060-f002:**
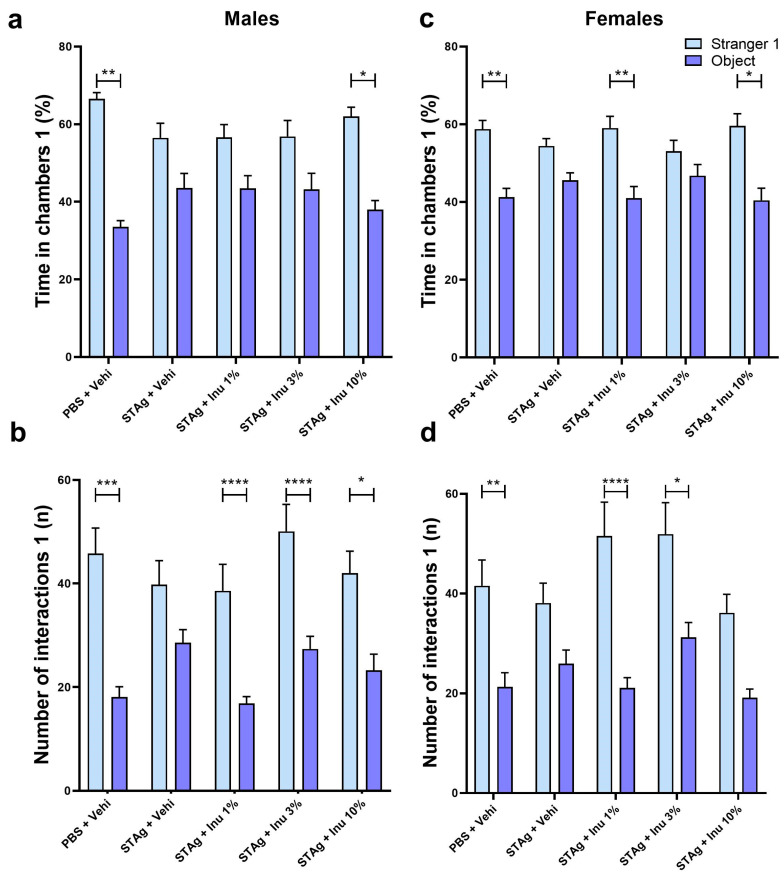
Social interaction test males and females 1. (**a**,**c**) Time in chambers (%). (**b**,**d**) Interactions (n). Bars represent M ± SEM. Inu = inulin; STAg = soluble Toxoplasma gondii antigen; PBS = phosphate-buffered saline. (*) indicates *p* < 0.05; (**) indicates *p* < 0.01; (***) indicates *p* < 0.001; and (****) indicates *p* < 0.0001 by Sidak’s test. N = 8–14.

**Figure 3 microorganisms-14-00060-f003:**
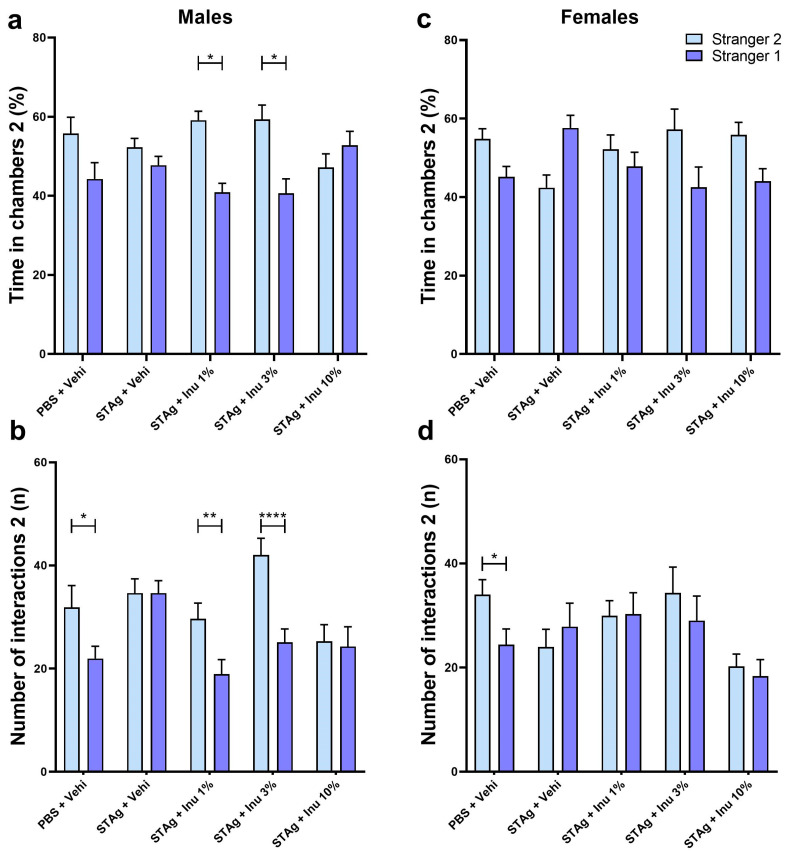
Social novelty in the social interaction test for male and female offspring. (**a**,**c**) Time in each chamber (%). (**b**,**d**) Interactions performed (n). Bars represent M ± SEM. Inu = inulin; STAg = soluble Toxoplasma gondii antigen; PBS = phosphate-buffered saline. (*) indicates *p* < 0.05; (**) indicates *p* < 0.01; and (****) indicates *p* < 0.0001 by Sidak’s test. N = 8–14.

**Figure 4 microorganisms-14-00060-f004:**
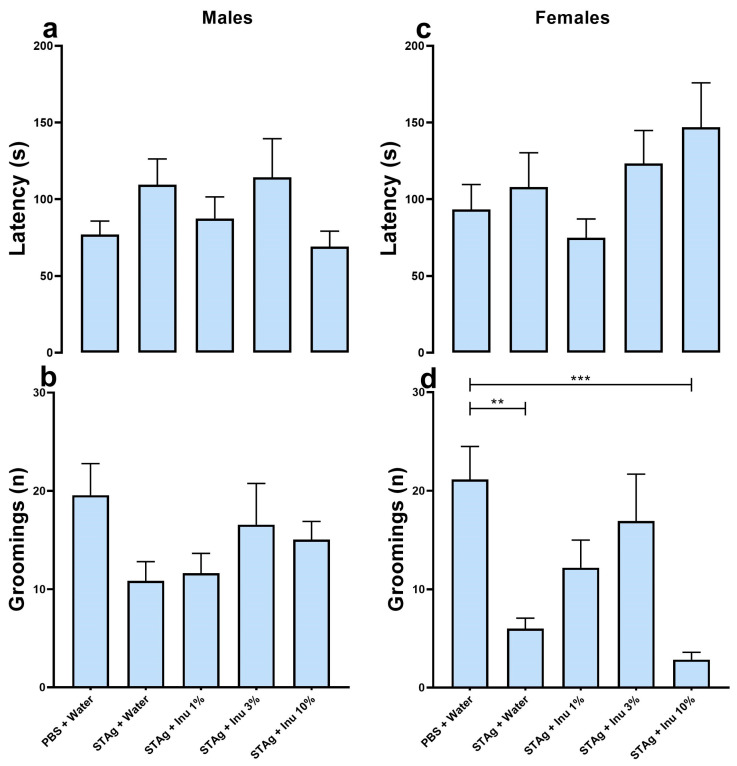
Sucrose spray test. (**a**,**b**) Males. (**c**,**d**) Females. a/c = latency to the first facial grooming (s). b/d = number of facial groomings performed (n). Bars represent M ± SEM. Inu = inulin; STAg = soluble Toxoplasma gondii antigen; PBS = phosphate-buffered saline. (**) indicates *p* < 0.01 and (***) indicates *p* < 0.001 by Dunn’s test. N = 7–21.

**Figure 5 microorganisms-14-00060-f005:**
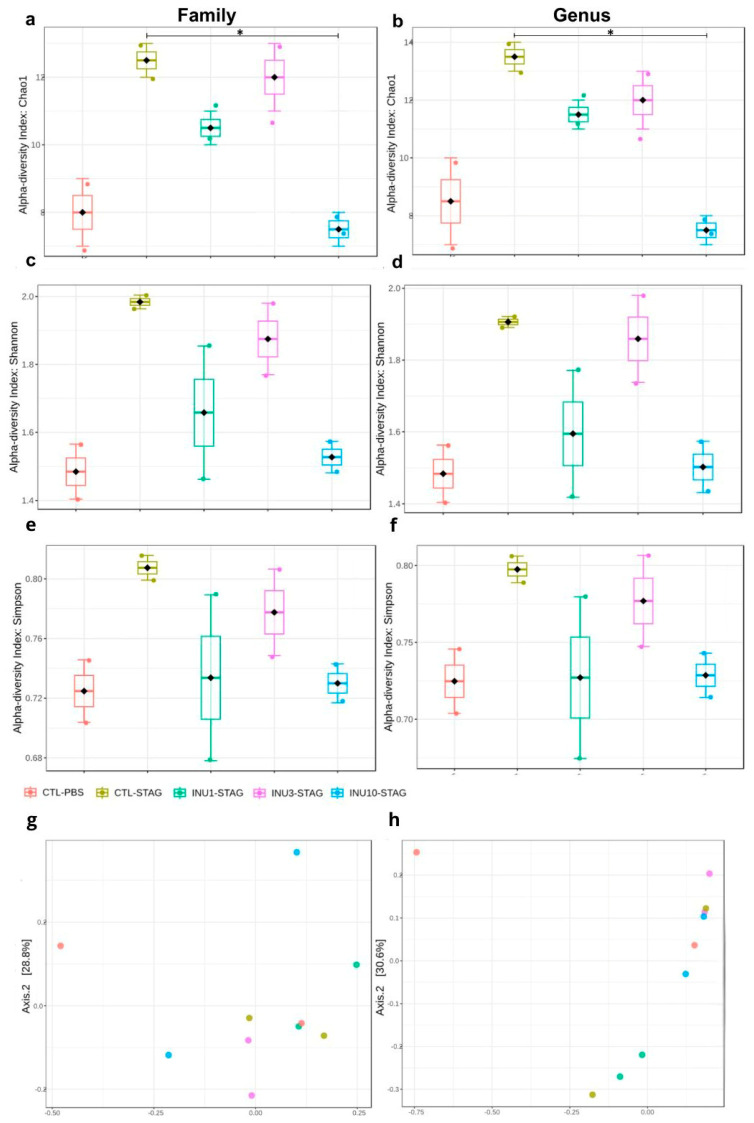
Effect of STAg and inulin treatment on the composition of fecal microbiota in the offspring. (**a**,**b**) Alpha diversity according to the Chao1 index. (**c**,**d**) Alpha diversity according to the Shannon index. (**e**,**f**) Alpha diversity according to Simpson index. (**g**,**h**) Beta diversity expressed in the Principal Coordinate Analysis (PCoA) plot with Bray–Curtis dissimilarity. Groups are represented as follows: CTL-PBS (red), control + PBS; CTL-STAg (yellow), control + STAg; INU1-STAg (green), inulin 1% + STAg; INU3-STAg (purple), inulin 3% + STAg; and INU10-STAg (blue), inulin 10% + STAg. (*) indicates *p* < 0.05.

**Figure 6 microorganisms-14-00060-f006:**
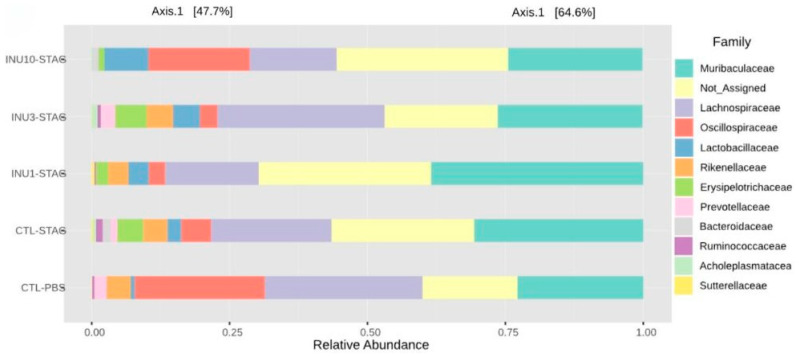
Stacked bar graphs showing true relative abundance of bacterial genera within groups.

**Figure 7 microorganisms-14-00060-f007:**
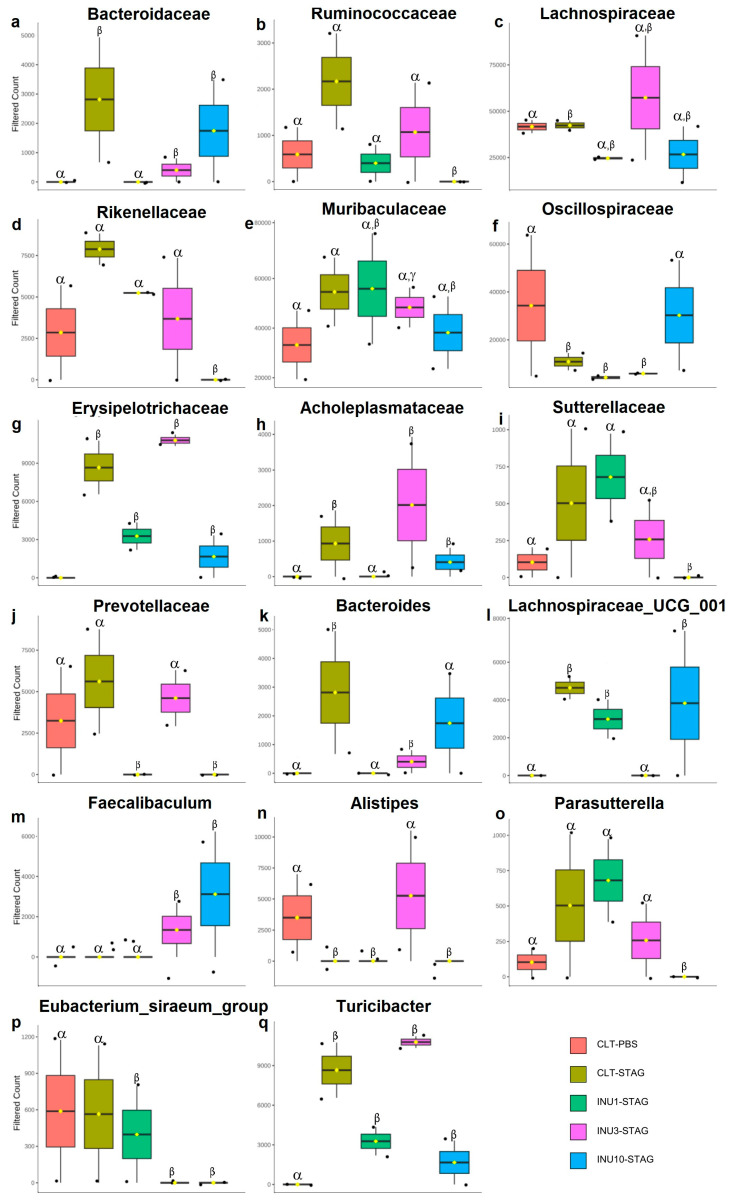
Impact of maternal STAg and inulin treatment on the relative abundance of offspring gut microbiota at the family and genus levels. (**a**–**j**) Family level. (**k**–**q**) Genus level. The Greek letters alpha, beta, and gamma represent the experimental groups. Bars with different letters are statistically different (*p* < 0.0001). Groups are represented as follows: CTL-PBS (red), control + PBS; CTL-STAg (yellow), control + STAg; INU1-STAg (green), inulin 1% + STAg; INU3-STAg (purple), inulin 3% + STAg; and INU10-STAg (blue), inulin 10% + STAg. The dots represent individual data, and the bars represent the mean and standard deviation (SD).

## Data Availability

The data that support the findings of this study are openly available in Zenodo at https://doi.org/10.5281/zenodo.14278547, reference number 10.5281/zenodo.14278547.

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
