# Peer review of "Prebiotic Treatment During Pregnancy Prevents Social Deficits Associated with Autism Spectrum Disorder-like Behavior Induced by Maternal Immune Activation"

_microorganisms, 2025, doi:10.3390/microorganisms14010060_

Round 1

Reviewer 1 Report

Comments and Suggestions for Authors

Comments and suggestions

  1. The manuscript provides no power calculations or a priori criteria for the numbers of dams and offspring analyzed, undermining confidence in negative findings and detected effects. Include a power analysis based on the primary outcome to justify group size.
  2. Offspring from the same dam are not independent, yet each animal is treated as an experimental unit. Model litter as a random effect or average data per litter to avoid pseudoreplication.
  3. Dams are said to be “randomly assigned,” but the procedure (random-number generator, blocking, etc.) is not described. Explaining the allocation strategy and who performed it enhances reproducibility.
  4. Behavioural scoring was blinded, but it is unclear whether microbiome processing and statistical analyses were performed blind to group allocation. Describe blinding procedures for every stage.
  5. Although both sexes are included, some analyses pool sexes or omit sex × treatment interactions. Use full factorial models or separate analyses with justification.
  6. The 1 %, 3 % and 10 % inulin concentrations lack justification regarding maternal tolerance or expected butyrate levels. Provide pharmacokinetic references or pilot data supporting these doses.
  7. Samples are pools of male offspring and rely on forward reads only, limiting taxonomic resolution. Sequence individual samples from both sexes with full 2 × 250 bp reads to strengthen diversity analyses.
  8. Microbiome was collected only at P55; early postnatal dynamics remain unexplored. Implement longitudinal sampling (P21, P35, P55) to link microbial shifts with behavioural development.
  9. The dose is expressed as 30 µg/L rather than per animal; clarify the delivered dose, antigen purity and endotoxin levels. Include LPS controls to rule out non-specific effects.
  10. The panel focuses on social interaction, EPM, sucrose spray and marble burying, but lacks dedicated motor-stereotypy and communication (ultrasonic vocalization) assays. Adding them would improve construct validity for ASD-like phenotypes.

Author Response

Comments and suggestions  - Reviewer 1

  1. The manuscript provides no power calculations or a priori criteria for the numbers of dams and offspring analyzed, undermining confidence in negative findings and detected effects. Include a power analysis based on the primary outcome to justify group size.

Thank you for pointing this out. We agree with this comment, and included the power analysis as requested.

"To achieve 80% power at a two-sided α=0.05 to detect a between-group difference of Δ=20 units (1 SD). Because offspring within a litter are correlated, the litter (dam) was treated as the experimental unit. Considering m = 6 offspring measured per litter and an intra-litter intraclass correlation (ICC) of 0.40, yielding a design effect DE = 1 + (m−1)×ICC = 3. Under these assumptions and SD=20, the required number of litters per group was 8.”

  1. Offspring from the same dam are not independent, yet each animal is treated as an experimental unit. Model litter as a random effect or average data per litter to avoid pseudoreplication.

Thank you dor pointing this out. This is a good point and contributed for the analysis. We agree that offspring from the same dam are not independent. In our analyses, the dam (litter) was the experimental unit. Specifically, we fit linear mixed-effects models with a random intercept for litter to account for intra-litter correlation. This approach avoids pseudoreplication and aligns the analysis with the experimental unit.

  1. Dams are said to be randomly assigned,” but the procedure (random-number generator, blocking, etc.) is not described. Explaining the allocation strategy and who performed it enhances reproducibility.

Thank you for pointing this out. We have described the method used to blind and randomly assign the experimental groups.

"After a one-week acclimation period at the experimental bioterium, an animal facility assistant, blinded to the experimental procedures, randomly placed the males and females mated at a ratio of 1:3 until the observation of a vaginal plug, which was considered as gestation day 0.5. Females were isolated, and their weight was monitored daily to track gestational progression. Upon confirmation of pregnancy, the assistant isolated the dams to be randomly assigned to the experimental groups (5–6 animals per group)."

  1. Behavioural scoring was blinded, but it is unclear whether microbiome processing and statistical analyses were performed blind to group allocation. Describe blinding procedures for every stage.

Thank you for pointing this out. We have specified the method adopted to blind the analysis, as described below.

"For the 16s sequencing and microbiome analyst, each fecal sample ware codified by number, and processed by an experimenter blinded to the group. The code was revealed only after sequencing, before upload to the MicrobiomeAnalyst platform.”

  1. Although both sexes are included, some analyses pool sexes or omit sex × treatment interactions. Use full factorial models or separate analyses with justification.

Thank you for pointing this out. The referee ar right with this concern. Initially, we had made analysis using full factorial models by both method, ANOVA and generalists models (GLM) setting sex as a factor. However, non effect was observed or interaction for sex and treatments. For this reason, we decided do perform male and female analysis separately. We have already included this concern at the Limitation section.

  1. The 1 %, 3 % and 10 % inulin concentrations lack justification regarding maternal tolerance or expected butyrate levels. Provide pharmacokinetic references or pilot data supporting these doses.

Thank you for pointing this out. The referee is correct in this regard. Previous studies have used a broad range of inulin doses and treatment schemes, most of them ranging from 2% to 10% administered through diet supplementation (see Yu, Y., He, J., et al., 2025; doi.org/10.1111/1750-3841.70250 for details). For the rationale of the present study, we based in two works that used inulin as a prebiotic or synbiotic formulation at doses around 5% in diet supplementation or 2 g/kg/bw. These studies reported either a reduction of neurotoxicity in offspring from dams exposed to the rotenone model (Krishna, G., Muradidhara, 2018; DOI: 10.1016/j.biopha.2018.05.107), or protective effects on social behaviors in the VPA/ASD model, accompanied by increased fecal levels of L. reuteri (Wang, C., et al., 2024; DOI:10.1039/d3fo02663a). Both studies reported good tolerability of inulin. To adjust for our laboratory conditions, we prioritized establishing a dose–response curve of inulin treatment (log10). We have included this information at text body to clarify as suggested by the referee.

  1. Samples are pools of male offspring and rely on forward reads only, limiting taxonomic resolution. Sequence individual samples from both sexes with full 2 × 250 bp reads to strengthen diversity analyses.

Thank you for pointing this out. This is an interesting point. The referee’s suggestion would enhance our ability to identify strengths and improve the analysis. This limitation was also mentioned in the Limitations section.

  1. Microbiome was collected only at P55; early postnatal dynamics remain unexplored. Implement longitudinal sampling (P21, P35, P55) to link microbial shifts with behavioural development.

Thank you for pointing this out. This is an interesting point. Indeed, the ecological evolution of the enteric microbiome is a dynamic process that begins at birth and continues until death (although this is debated by some authors), with major modifications occurring during the first three years of life—the (first thousand days) in humans (Opstal, E.J., & Bordenstein, S., 2015; DOI: 10.1126/science.aab3958; Martino, C., et al., 2022; DOI: 10.1038/s41579-022-00768-z; Naspolini, N.F., et al., 2024; DOI: 10.3390/microorganisms12030424). Therefor, the referee is right in question regards the exploration of the microbiome during the others point, considering that previous studies from our group and others have shown that disturbances in the microbiome during these early stages may be associated with psychiatric disorders in adult life (Hassib, L., et al., 2023; Hassib, L., et al., 2025; Ma, Z.M., et al., 2024; DOI: 10.1038/s41392-024-01946-6). However, considering that the protective effect of inulin against STAg antigens was still unknown, and that we performed a dose–response curve to explore this effect, our strategy prioritized analyzing the endpoint at which behavioral abnormalities associated with ASD, anxiety, and MDD could be observed. After identifying the most effective dose of inulin, a further study to determine the time point at which the gut microbiome is most affected may be performed.

  1. The dose is expressed as 30 µg/L rather than per animal; clarify the delivered dose, antigen purity and endotoxin levels. Include LPS controls to rule out non-specific effects.

Thank you for this correction. We have updated the dose of the STAg antigen in the text as described below. Regarding the LPS controls, previous studies from our group have shown that the behavioral impairments associated with MIA may depend on the origin of the antigens (H1N1, LPS, or STAg) and the neurodevelopmental stage at which the maternal immune challenge occurs (Spini, V.B.M.G. et al., 2020). Therefore, we chose to use PBS as the control, since the purified STAg is diluted in this buffer.

"On E16.5, pregnant female mice were subcutaneous injected with a single dose of 0.06 mg/Kg of STAg solution at concentration (30 mg/L) or PBS, s.c. injection in a volume of 2 µL/g animal,”.

  1. The panel focuses on social interaction, EPM, sucrose spray and marble burying, but lacks dedicated motor-stereotypy and communication (ultrasonic vocalization) assays. Adding them would improve construct validity for ASD-like phenotypes.

Thank you for this advice. Indeed, the inclusion of ultrasonic vocalization would add another parameter to the evaluation of social behavior impairments or protection associated with MIA/prebiotic maternal treatment, and would allow the assessment of social behavior in young mice. However, the three chamber method is an improvement of arena apparatus and has been applied to access sociability and empathy/novelty preference behavior in rodent models. Regarding stereotypical behavior, we prioritized its evaluation through marble burrowing, and, indirectly, by quantifying the number of arm crossings in the EPM task, which may provide clues about stereotypy or motor abnormalities. However, the referee is correct that a task specifically designed to assess motor stereotypy would improve the evaluation. This approach could be incorporated into future studies.

Reviewer 2 Report

Comments and Suggestions for Authors

This is a well-designed and properly executed series of animal experiments aimed at investigating how maternal immune activation and its prebiotic treatment affect the offspring's microbiome composition as well as their social and behavioral abilities.

For better and easier understanding, I would like to suggest the following minor changes.

  • Methods: Figure 1 should be separated into two figures. In the revised Figure 1, alongside the experimental design, an illustrative representation of each behavioral test should be included (3.3.1, 3.3.2, 3.3.3, 3.3.4) in an adequate size. Figure 1b and 1c should be presented as a new, separate figure and relocated to Section 4 (Results).
  • Please indicate in Sections 3.3.1, 3.3.2, 3.3.3, and 3.3.4 the interpretation associated with each behavioral test.
  • "The subsections of Section 4 (Results) should follow the same sequence as outlined in subsection 3.3 (Behavioral Tests)
  • The titles of Sections 4.1, 4.2, 4.3, and 4.4 should include the name of the behavioral test from which the respective results were obtained.
  • In Figures 1b and 1c, 2a and 2c, and 3a and 3c, please include and clearly label the x-axis.
  • Figures 4 and 5 should be moved to the supplementary files, as they do not show any statistically significant differences.
  • The subfigures (a–i) in Figure 6 are currently too small to be easily interpreted. Please divide Figure 6 into two separate figures for improved readability. Revised Figure 6 should include enlarged versions of subfigures a–h. A new figure (e.g., Figure 7) should present subfigure 6i.
  • Lines 444-451: An altered alpha diversity should not be directly interpreted as a disruption of the gut microbiota, please rephrase.
  • Based on your findings (lines 475–479), there appears to be a contradiction in the discussion section (lines 527–530).
  • Discussion section: please provide comments on the challenges associated with comparing microbiome data obtained at different taxonomic levels, such as phylum, family, and species.

Author Response

Comments and suggestions  - RREVIEWER 2

This is a well-designed and properly executed series of animal experiments aimed at investigating how maternal immune activation and its prebiotic treatment affect the offspring's microbiome composition as well as their social and behavioral abilities.

For better and easier understanding, I would like to suggest the following minor changes.

  • Methods: Figure 1 should be separated into two figures. In the revised Figure 1, alongside the experimental design, an illustrative representation of each behavioral test should be included (3.3.1, 3.3.2, 3.3.3, 3.3.4) in an adequate size. Figure 1b and 1c should be presented as a new, separate figure and relocated to Section 4 (Results).

Thank you for pointing this out. We proceeded as recommended.

  • Please indicate in Sections 3.3.1, 3.3.2, 3.3.3, and 3.3.4 the interpretation associated with each behavioral test.

Thank you for pointing this out. The interpretation for each behavioral task was included as described below:

“The task is carried out in two stages. In the first stage, the animal’s preference is evaluated by measuring the time spent or exploratory activity (interactions) with another animal compared to an object. In the second stage, called the novelty sociability test, the evaluation focuses on whether the animal prefers to explore an unfamiliar animal over a familiar one.”

"The reduction of exploratory activity in the open arms (unprotected areas) or an increased number of risk-assessment behaviors may be interpreted as generalized anxiogenic-like behavior.”

"Animals that do not respond to the sweet stimulus of a sucrose spray applied dorsally may be interpreted as displaying anhedonic-like behavior, similar to the loss of interest in previously pleasurable activities observed in patients with major depressive disorder.”

"An increased number of marbles buried is interpreted as repetitive/compulsive-like and stereotyped-like behavior.”.

  • "The subsections of Section 4 (Results) should follow the same sequence as outlined in subsection 3.3 (Behavioral Tests);

Thank you for pointing this out. We modify the method section sequence to adjust to the result subsections.

  • The titles of Sections 4.1, 4.2, 4.3, and 4.4 should include the name of the behavioral test from which the respective results were obtained.

Thank you for pointing this out. The names of the tests were included as suggested.

  • In Figures 1b and 1c, 2a and 2c, and 3a and 3c, please include and clearly label the x-axis.

Thank you for pointing this out. We have included the label to x-axis as suggested.

  • Figures 4 and 5 should be moved to the supplementary files, as they do not show any statistically significant differences.

Thank you for pointing this out. We proceeded as recommended.

  • The subfigures (a–i) in Figure 6 are currently too small to be easily interpreted. Please divide Figure 6 into two separate figures for improved readability. Revised Figure 6 should include enlarged versions of subfigures a–h. A new figure (e.g., Figure 7) should present subfigure 6i.

Thank you for pointing this out. We proceeded as recommended.

  • Lines 444-451: An altered alpha diversity should not be directly interpreted as a disruption of the gut microbiota, please rephrase.

Thank you for pointing this out. The referee is right on this observation. We rephrase to term “disruption" to “modulation" proceeded as recommended.

"These results together indicate that the behavioral changes observed in adult offspring related to ASD, resulting from maternal immunogenic stress during pregnancy, may stem from gut microbiota modulation, which is passed from mother to offspring.”

  • Based on your findings (lines 475–479), there appears to be a contradiction in the discussion section (lines 527–530).

Thank you for pointing this out. The referee is correct, and we have rephrased, as shown below, to adopt a more conservative approach.

"Finally, the effects of STAg exposure may be associated with social deficits. Its effects on anxiety-like, depressive-like, or obsessive-compulsive-like behaviors would warrant further analysis.”

  • Discussion section: please provide comments on the challenges associated with comparing microbiome data obtained at different taxonomic levels, such as phylum, family, and species.

Thank you for pointing this out. The referee is correct, and we have included the paragraph below to highlight this concern.

"However, to draw biological inferences using microbiome panel from different taxonomic levels such as phylum, family, and genus is challenging task and must concern about the variability and lower taxonomic resolution like species. These discrepancies can lead to difficulties in comparing findings across studies that utilize different taxonomic resolutions. Moreover, strain-level diversity within a species can impact microbiome functionality, complicating comparisons between studies that may not resolve taxa to the strain level."
